# Brain-Derived Neurotrophic Factor Is Indispensable to Continence Recovery after a Dual Nerve and Muscle Childbirth Injury Model

**DOI:** 10.3390/ijms24054998

**Published:** 2023-03-05

**Authors:** Brian M. Balog, Kangli Deng, Tessa Askew, Brett Hanzlicek, Mei Kuang, Margot S. Damaser

**Affiliations:** 1Department of Biomedical Engineering, Lerner Research Institute, Cleveland Clinic, Cleveland, OH 44195, USA; 2Advanced Platform Technology Center, Research Service, Louis Stokes Veterans Affairs Medical Center, Cleveland, OH 44106, USA; 3Department of Biology, University of Akron, Akron, OH 44325, USA; 4Department of Biology, Case Western Reserve University, Cleveland, OH 44106, USA; 5Glickman Urologic and Kidney Institute, Cleveland Clinic, Cleveland, OH 44311, USA

**Keywords:** neuromuscular junction, TrkB, urinary incontinence, female, reinnervation

## Abstract

In women, stress urinary incontinence (SUI), leakage of urine from increased abdominal pressure, is correlated with pudendal nerve (PN) injury during childbirth. Expression of brain-derived neurotrophic factor (BDNF) is dysregulated in a dual nerve and muscle injury model of childbirth. We aimed to use tyrosine kinase B (TrkB), the receptor of BDNF, to bind free BDNF and inhibit spontaneous regeneration in a rat model of SUI. We hypothesized that BDNF is essential for functional recovery from the dual nerve and muscle injuries that can lead to SUI. Female Sprague–Dawley rats underwent PN crush (PNC) and vaginal distension (VD) and were implanted with osmotic pumps containing saline (Injury) or TrkB (Injury + TrkB). Sham Injury rats received sham PNC + VD. Six weeks after injury, animals underwent leak-point-pressure (LPP) testing with simultaneous external urethral sphincter (EUS) electromyography recording. The urethra was dissected for histology and immunofluorescence. LPP after injury and TrkB was significantly decreased compared to Injury rats. TrkB treatment inhibited reinnervation of neuromuscular junctions in the EUS and promoted atrophy of the EUS. These results demonstrate that BDNF is essential to neuroregeneration and reinnervation of the EUS. Treatments aimed at increasing BDNF periurethrally could promote neuroregeneration to treat SUI.

## 1. Introduction

Stress urinary incontinence (SUI), the leakage of urine due to increased abdominal pressure, is common among elderly women and diminishes quality of life [1]. During vaginal delivery, the baby’s head can injure the maternal pudendal nerve (PN) and external urethral sphincter (EUS) while passing through the birth canal [2]. Up to 40% of women suffer from post-partum SUI after pregnancy; these women are 2.4 times more likely to develop SUI later in life, suggesting a correlation between post-partum incontinence and SUI later in life [3]. An improved understanding of the pathophysiology could reveal targets for therapies.

Rodent models of SUI have demonstrated that a combined pudendal nerve crush (PNC) and vaginal distension (VD) injury results in decreased leak-point pressure (LPP) and SUI, and doubles recovery time compared to either injury alone [4]. PNC + VD also delays PN motor-function recovery, even though VD does not impair PN function [4]. Pan et al. demonstrated that brain-derived neurotrophic factor (BDNF) is dysregulated after PNC + VD, suggesting that BDNF is important for regeneration of the pudendal nerve and re-establishment of continence [5]. Gill et al. showed accelerated regeneration and functional recovery after PNC + VD injury with BDNF treatment [6]. In addition, Yuan et al. demonstrated that BDNF is needed for promotion of regeneration with stem cell secretome treatment [7]. However, it is unknown if BDNF is necessary for spontaneous functional recovery after this dual nerve and muscle injury.

Inhibiting the BDNF pathway in vivo is difficult, since BDNF knockouts are lethal [8]. Using a conditional BDNF knockout mice or heterozygous BDNF mouse models is not reliable, as a recent study has challenged the validity of mouse VD models [9]. A tyrosine kinase B (TrkB) fusion chimera (TrkB-Fc) has been previously shown to inhibit the BDNF regeneration pathway in spinal-cord-injury models at a dose of 12 μg per day [10,11]. Additionally, this TrkB-Fc has been shown to bind free BDNF and inhibit the BDNF pathway after PNC [12]. Using TrkB, the aim of this study was to test the hypothesis that BDNF is necessary for spontaneous functional recovery after the dual nerve and muscle injuries that can lead to SUI.

## 2. Results

### 2.1. LPP and Electrophysiological Results

LPP measured 6 weeks after the PNC + VD injury was not significantly different from that of sham-injured rats (Figure 1 and Figure 2). In contrast, LPP was significantly decreased 6 weeks after PNC +VD injury treated with TrkB compared to LPP after PNC + VD alone (*p* = 0.031), indicating that TrkB treatment inhibits recovery of LPP after PNC + VD. There were no significant differences in EUS EMG firing rate or amplitude between the Sham Injury group and either the PNC + VD Injury group or the Injury + TrkB group (Figure 3). Although EUS EMG frequency and amplitude in Injury + TrkB rats appeared to be less than those after PNC and VD injury, these differences were not statistically significant.

### 2.2. BDNF and NT4 Plasma Concentration

BDNF plasma concentrations 6 weeks after PNC + VD were not significantly different between Injury and Injury + TrkB groups (Figure 4A). In contrast, NT4 plasma concentrations were significantly increased in the Injury + TrkB group compared to Injury alone 6 weeks after injury (*p* = 0.049; Figure 4B).

### 2.3. EUS Morphology and NMJs Staining

EUS morphology of Sham Injured rats showed an intact EUS with little collagen infiltration, and the EUS had some collagen infiltration 6 weeks after injury (Figure 5). In contrast, 6 weeks after injury and TrkB, the EUS was not intact, and all rats demonstrated collagen infiltration between striated muscle fibers of the EUS (Figure 5). NMJ morphology showed that the Sham Injury group had intact NMJs innervated by a single axon, and 6 weeks after PNC + VD, the NMJs were intact, some of which were innervated by multiple axons. Six weeks after injury + TrkB, there were fewer innervated NMJs than after injury alone (Figure 5).

## 3. Discussion

While most women recover continence one year after childbirth, many will have reoccurrence within five years [13,14]. Women who suffer from post-partum SUI are also 2.4 times more likely to develop SUI later in life [15], suggesting that tissue regeneration after the maternal injuries of childbirth is insufficient and can lead to later development of SUI [16,17]. Animal models of SUI have shown that BDNF expression is dysregulated in the EUS after PNC + VD injury [18]. The goal of this study was to test the hypothesis that BDNF is necessary for functional recovery after PNC + VD. We have previously demonstrated that TrkB-Fc treatment inhibits continence recovery after PNC, so this method of inhibiting BDNF was utilized in the current study at the same dose [12].

We demonstrated no significant differences in LPP and EUS EMG between the Sham Injury and Injury groups, showing that continence recovered within six weeks after PNC + VD, as has been demonstrated previously [4,19]. LPP was chosen as the primary outcome of the study, as it is a key indicator of SUI [20]. Additionally, the continence mechanism has been shown to have similar contributors (i.e., primarily EUS and urethral smooth muscle) in both humans and rats [21,22].

LPP was significantly decreased after injury and TrkB compared to injury alone, indicating that TrkB inhibited recovery and BDNF is necessary for spontaneous recovery of continence after injury. These results are consistent with previous studies that demonstrate that TrkB-Fc administration inhibits functional recovery from nerve injury [10]. Li et al. showed in a rat model of spinal-cord injury that TrkB treatment inhibited accelerated functional recovery via treadmill training, which is also thought to occur via a BDNF-mediated mechanism [10].

EUS EMG amplitude after injury and TrkB decreased compared to that of the Injury group and trended toward significance (*p* = 0.06), supporting the LPP outcomes. LPP was the primary outcome of the study, and a sample size analysis indicated that 13 animals per group would be needed to power the analysis. Secondary outcomes, such as EUS EMG, had higher standard errors, resulting in a lack of significant differences between groups. Additionally, the EUS is only one contributor of several to the continence mechanism, which could also explain why we did not see a significant difference in EUS results while seeing a significant difference in LPP [22]. In support of this idea, Dissaranan et al. found that after stem cell treatment, LPP was significantly increased compared to saline treated animals, while not demonstrating a significant increase in EUS EMG outcomes [23]. Likewise, Yuan et al. found that reducing BDNF in stem cell secretome treatment reduced LPP recovery after PNC + VD but resulted in no significant difference in EMG outcomes [7]. The EUS EMG amplitude trend is also in agreement with the results of Byrne et al., who showed a decrease in muscle compound action potential after facial nerve transection treated with an anti-BDNF antibody [24].

The Masson-trichrome-stained specimens supported the recovery of continence 6 weeks after injury, since the morphology results were similar for the Sham Injury group and the Injury group. However, qualitative EUS morphology appeared more disrupted in the Injury + TrkB group than the Injury group. NMJ analysis supports these results, since fewer NMJs were innervated in the Injury + TrkB group compared to the Injury group. Decreased recovery has been associated previously with a smaller number of intact EUS fibers, decreased NMJ innervation, and increased collagen infiltration of the EUS [25].

While a decrease in functional recovery was detected, a reduction in BDNF plasma concentration was not detected in the dual Injury + TrkB group after 6 weeks of treatment. In contrast, previous studies have shown a significant decrease in free BDNF (~45%) with 3 weeks of TrkB treatment, while showing a significant decrease in PN function [12]. Additionally, one week of treatment with TrkB significantly decreased expression of the regeneration-associated gene βII tubulin, which previously has been shown to be key indicator of regeneration [26]. The results of this study, along with those of Balog et al., suggest that by six weeks after dual injury, a compensatory mechanism was increasing BDNF levels, which was suggested by the increase in variability in the BDNF plasma concentration. Additionally, researchers have shown that a reduction in BDNF by 25% in culture has a detrimental effect on dendrite growth, suggesting a decrease of 45% is sufficient to inhibit the regeneration pathway, even if it is only for the first three weeks. Nonetheless, no hypothesized compensatory mechanism was able to improve functional outcomes.

Supporting this idea is the significant increase in NT4 plasma concentration in the Injury + TrkB group. TrkB-Fc was previously shown not to significantly change the NT4 plasma concentration with 3 weeks of treatment [12]. In contrast, we observed a significant increase in NT4 concentration with 6 weeks of treatment in this study. Gill et al. demonstrated a compensatory effect when treating the dual nerve and muscle injury model with BDNF, in which BDNF expression was decreased in the EUS [12]. Additionally, Moffat et al. found that inhibiting vascular endothelial growth factor (VEGF)-A expression caused upregulation of VEGF-D expression, suggesting that inhibiting one protein causes a biological system to upregulate related genes in response [27]. Therefore, the same compensatory mechanism could increase the expression of both ligands (BDNF and NT4) that bind to TrkB.

In this study, we did not determine if the remaining TrkB-Fc was functionally active after six weeks in the pump. The pumps used in this study were designed to last 44 days, resulting in a small amount of residual volume (~0.0047 mL) being collected after six weeks of use. Thus, while we know from previous studies that the TrkB-Fc molecule was functional at 3 weeks, we cannot be sure it was functional in weeks 3–6. Additionally, although we chose our dose of TrkB-Fc because it had previously been used safely in vivo, a larger dose may have demonstrated a greater effect [10,28]. However, this would require testing multiple doses to ensure safety in vivo, which was outside the scope of this study. Additionally, since we only tested functional recovery at 6 weeks, it is possible that functional recovery was delayed, rather than inhibited, as it could potentially happen at a later time. However, previous studies have shown that functional recovery occurs by 6 weeks after injury, suggesting that in this study, we inhibited the normal regenerative process [4,19].

The results of this study, when examined with the results of Gill et al., support the idea that BDNF is both necessary and sufficient for continence regeneration after a dual nerve and muscle childbirth injury [6]. Gill et al. demonstrated that BDNF administration accelerated LPP recovery 2 weeks after injury [6]. Additionally, Masson’s staining showed EUS morphology in BDNF-treated animals similar to that of sham animals. While the current study showed that TrkB administration inhibited LPP recovery, EUS morphology in TrkB-treated animals showed fewer intact fibers than the saline-treated animals 6 weeks after injury, supporting the functional outcomes.

The regenerative capacity of neurons, Schwann cells, and muscle cells decreases over time, as indicated by decreased expression level of regeneration-associated genes [29,30]. This suggests that the longer it takes for the pudendal nerve to regenerate, the less likely it is to encounter a growth-permissive environment, leading to fewer axons reinnervating the EUS, and resulting in more EUS muscle fibers being innervated by a single neuron, as depicted in Figure 6. Continence would be restored in this scenario, but this limited regeneration paradigm could explain the development of SUI decades later—that is, the loss of innervation to muscles with ageing leading to muscle weakness and redevelopment of SUI [31]. Song et al. showed functional recovery with impaired NMJ recovery 9 weeks after a PNC + VD dual injury, suggesting this impaired situation [32]. Insufficient regeneration would not only be compounded by aging but also by other risk factors for SUI, such as smoking, obesity, and multiple vaginal deliveries [3,33].

## 4. Materials and Methods

The Louis Stokes Veterans Affairs Medical Center Institutional Animal Care and Use Committee (IACUC) approved this study. Thirty-seven virgin female Sprague–Dawley rats (250–275 g) were divided into three groups: PNC + VD receiving saline (injury; *n* = 14), PNC + VD receiving TrkB-Fc (injury + TrkB; *n* = 11), and sham PNC + VD receiving saline (sham injury; *n* = 12). In vivo functional testing using leak-point pressure testing (LPP) as the same time as EUS electromyography (EMG) was performed (4–6 times) on the animals six weeks after the injury, a time point selected to ensure spontaneous recovery of the injury group based on prior research [19]. After functional testing, blood and tissues were harvested and stored for assessment of BDNF and Neurotrophin 4 (NT4) and histology and immunofluorescence, respectively.

### 4.1. Osmotic Pump Preparation

Osmotic pumps were prepared 40 h before the surgery (model 4004; Alzet, Cupertino, CA, USA). The pumps were filled with either saline or a TrkB-Fc/saline solution such that the animals received 12 μg TrkB-Fc (688-TK; R&D systems; Minneapoliss, MN, USA) per day, as we have done previously [12]. Pump weights were recorded before and after filling. After filling, the regulator was inserted, and the vinyl catheter was attached to the regulator. The pump was then placed in sterile saline in a 37 °C incubator until implantation. The dosage was chosen based on previous publications that used TrkB-Fc [10,11].

### 4.2. Injury model and Implantation of the Osmotic Pump

The creation of the injury and sham injury models was performed as previously described [19]. In summary, rats were anesthetized with 2–3% isoflurane. Rimadyl (5 mg/kg) was administered subcutaneously at the beginning of the procedure and again on the first postoperative day. The dorsal lumbar–sacral region was shaved and disinfected, and a midline incision was made. The gluteus maximus was cut proximal to the vertebral column. The ischiorectal fossa was gently opened to visualize the pudendal nerve, which was then crushed twice for 30 s with Castroviejo forceps. A subcutaneous pocket was created caudal to the incision site where the osmotic pumps were implanted. The catheter was threaded to the ischiorectal fossa and secured to muscles adjacent to the pudendal nerve near the injury site. This process was then repeated on the contralateral side to ensure both pudendal nerves were crushed and treated. Sham PNC nerves were exposed but not crushed. The gluteus maximus muscle and skin were then closed separately.

While remaining anesthetized, animals were moved to the supine position, and the vagina was dilated with a series of lubricated urethral dilators (French 24–32). A modified Foley catheter was then placed inside the vagina and inflated with 3 mL water for four hours. The Foley catheter was then deflated and removed. During the procedure, the animals’ respiration rate was monitored, and isoflurane levels were adjusted to ensure that animals were not over anesthetized. Animals that received sham VD had the vaginal catheter inserted but not inflated. A second dose of rimadyl was given the next day.

### 4.3. Suprapubic Bladder Catheter and Functional Testing

Functional outcomes were evaluated as previously described under urethane (1.2 g/kg) anesthesia via intraperitoneal injection [34]. The animals were initially anesthetized with isoflurane (2–3%) during surgery and were then transferred to urethane anesthesia before functional testing. A midline incision was performed in the abdomen ~2 cm above the urethral meatus, and the bladder was exposed. After placing a purse-string suture in the dome of the bladder, an incision was performed at the center, and a catheter with a flared tip (PE-50 tubing) was placed in the bladder. The suture was then tightened around the catheter. After inspecting the bladder incision for leakage during a test filling, the abdominal muscle incision was sutured closed with a single stitch. The skin above the pubic symphysis was opened, and the muscle layer was dissected. The symphysis pubis was cut in the center to access the urethra. A pair of bipolar, parallel platinum–iridium electrodes were placed on the EUS for recording EUS EMG. The electrodes were connected to an amplifier (model PF11; AC Amplifier, Astro-Med; West Warwick, RI, USA; bandpass frequencies: 3 Hz–3 kHz), and recordings were performed using a Powerlab8/35 (ADInstruments, Inc., Colorado Springs, CO, USA; 10 kHz sampling rate). The bladder catheter was connected to a pressure transducer (Model PT300; Astro-Med.) and syringe pump (5 mL/hr), and the pressure data were amplified and recorded by the Powerlab system. For LPP testing, when the bladder was approximately half full, gentle pressure was applied to the abdomen, and the external pressure was continued until leakage was observed. The externally applied pressure was then rapidly removed. LPP and EUS EMG were recorded simultaneously and repeated 4–6 times per rat.

### 4.4. Plasma Collection

After functional testing, animals were euthanized by anesthetizing them with 5% isoflurane. After opening the chest cavity, blood was collected intracardiacally, placed in EDTA collection tubes, and stored at 4 °C until processing. Samples were centrifuged at 2500 rpm for 15 min at 4 °C. The plasma was collected and stored at −80 °C for later use.

### 4.5. Histology and Immunofluorescence

Urethras were flash frozen in OCT and stored at −20 °C. They were sectioned transversely at the EUS (7 and 14 µm thick), and slides were stored at −80 °C. Immunofluorescence of neuromuscular junctions (NMJs) and qualitative assessment were performed as previously described [12]. Innervation of the EUS was probed with anti-neurofilament 68 and 200 antibodies (1:400 dilution each; N0142 and N5139 Sigma-Aldrich, St. Louis, MI, USA), followed by a secondary antibody: Alexa Fluor 488 conjugated donkey anti-mouse IgG (1:400; Item No. R37114, ThermoFisher, Waltham, MA, USA). In total, 4 μg/mL of tetramethylrhodamine-conjugated alpha-bungarotoxin (1:400; Item No. T1175, ThermoFisher, Waltham, MA, USA) was used to identify NMJs, and Alexa Fluor 350 conjugated phalloidin (1:40; Item No. A22281, ThermoFisher, Waltham, MA, USA) was used to identify striated muscle of the EUS. Near sections were stained with Masson’s trichrome for morphological analysis.

Two blinded observers qualitatively evaluated representative histology and immunofluorescence images of each specimen, according to the following criteria: Masson’s-trichrome-stained urethral sections were used to assess EUS-striated muscle based on whether the EUS was intact and if collagen had infiltrated between the muscle fibers. EUS NMJs were assessed using immunofluorescence on whether NMJs were compact, innervated, and innervated by a single axon. Four to five animals were analyzed per group using one slide per animal, as we did previously [34,35].

### 4.6. ELISA

An BDNF ELISA assay (Item No. G7610, Promega, Madison, WI, USA) was performed on plasma, as described in the manufacturer’s instructions and as previously described [12]. In brief, the provided Anti-BDNF mouse antibody (1:1000) was coated on the 96-well plate overnight at 4 °C. The plate was then washed five times with 200 µL of tris-buffer saline with tween 20 (TBST) wash buffer. This step was repeated before applying new reagents to the plate throughout the protocol. The plate was then blocked for two hours with 1× blocking and sample buffer (provided in the kit). The plasma samples (1:4 dilution with 1× blocking buffer) and the BDNF standard curve (provided in the kit) were added to the plate and allowed to incubate for two hours at room temperature with shaking (400 rpm). The anti-human BDNF antibody (1:500; provided in the kit) was then applied and incubated for two hours while shaking. The plate was then incubated with an anti-IgY HRP conjugate (1:200; provided in the kit) for one hour with shaking. Then, 100 µL of TMB One solution (provided in the kit) was added to each well. The plate was then incubated for 10 min with shaking before the reaction was stopped by adding 100 µL of 1 N hydrochloric acid. The plate was then read on a plate reader.

Neurotrophin 4 ELISA (ERN0114; ABclonal, Woburn, MA, USA) was also performed on plasma as previously described [12]. In brief, samples and standards (provided in the kit) were added to the supplied plate with 50 µL of the supplied enzyme solution and then incubated for 1 h at 37 °C. The plate then washed five times with 400 µL of 1× wash solution. A 1:1 mixture of horseradish peroxidase (substance A 50 µL, provided in the kit) and TMB solution (substance B 50 µL, provided in the kit) was then added to each well and incubated in the dark for 15 min. Then, the stop solution (provided in the kit) was added to each well, and the plate was read on a plate reader. A Mann–Whitney test was used to indicate a statistically significant difference between the groups (*p* < 0.05).

### 4.7. Data Analysis

As performed previously, LPP was determined by subtracting baseline bladder pressure from peak bladder pressure during LPP testing [34]. Baseline pressure was defined as the bladder pressure just before the application of pressure, and peak pressure was the bladder pressure at which leakage occurred [36]. Analysis of LPP and EUS EMG signals was performed by sequestering 4–6 one-second segments for baseline and peak activity. The amplitude and firing rate of EUS EMG activity were determined as previously described [36]. In brief, a custom threshold for each baseline-peak pair was created to remove background noise. Methods were performed as previously described, using Matlab (V 2012b, Mathworks, Natick, MA, USA) [36]. The increases in amplitude and firing rate during LPP testing were determined by subtracting baseline from peak activity levels. The mean value for each animal for each quantitative variable was calculated and was used to calculate the mean and standard error for each experimental group. A Welsh one-way analysis of variance (ANOVA) with a Dunnett’s T3 post hoc test was used to compare LPP and EUS EMG outcomes between groups with the injury alone group as the control. *p* < 0.05 indicated a statistically significant difference between groups. Representative traces of the LPP pressure and corresponding EUS EMG trace were chosen based on means of both outcomes in each group (Figure 1).

## 5. Conclusions

In summary, BDNF is essential to functional recovery of the continence mechanism after a dual nerve and muscle injury, since TrkB inhibited or delayed spontaneous functional recovery. Masson’s and NMJ results support these findings, as TrkB treatment also inhibited anatomical recovery. The results of this study support the neurotrophic theory of SUI: that impaired neurotrophic expression results in impaired neuroregeneration and reinnervation of the EUS, suggesting a treatment: increased BDNF locally could be beneficial to women suffering from post-partum SUI and may prevent SUI later in life.

## Figures and Tables

**Figure 1 ijms-24-04998-f001:**
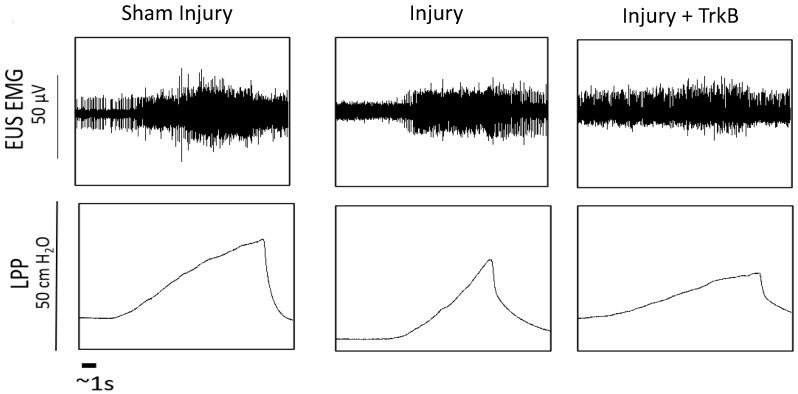
Examples of functional outcomes. Examples of external urethral sphincter electromyography (EUS EMG; **top**) and corresponding bladder pressure (**bottom**) from each of the three groups 6 weeks after injury: sham dual injury treated with saline (Sham Injury); pudendal nerve crush (PNC) and vaginal distension (VD) injury treated with saline (Injury); and PNC + VD injury treated with tyrosine kinase B (TrkB) fusion chimera (Injury + TrkB). Scale bar represents 1 s. Each animal was tested individually, and LPP and EUS EMG were recorded simultaneously. Representative examples were chosen by being close to the respective means for both outcomes.

**Figure 2 ijms-24-04998-f002:**
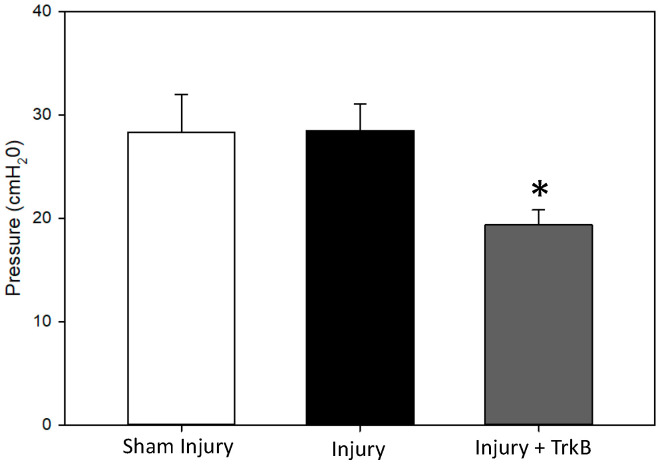
Leak-point pressure results. Leak-point pressure 6 weeks after sham injury treated with saline (Sham Injury); pudendal nerve crush (PNC) and vaginal distension (VD) injury treated with saline (Injury); or PNC + VD injury treated with tyrosine kinase B (TrkB) fusion chimera (Injury + TrkB). Each bar represents mean ± standard error of the data from 11–14 animals. There were 4–6 LPP tests per animal. A Welsh one-way analysis of variance (ANOVA) with a Dunnett’s T3 post hoc test was used to determine statistically significant differences (*p* < 0.05) compared to the Injury group. * indicates a statistically significant difference compared to the Injury group.

**Figure 3 ijms-24-04998-f003:**
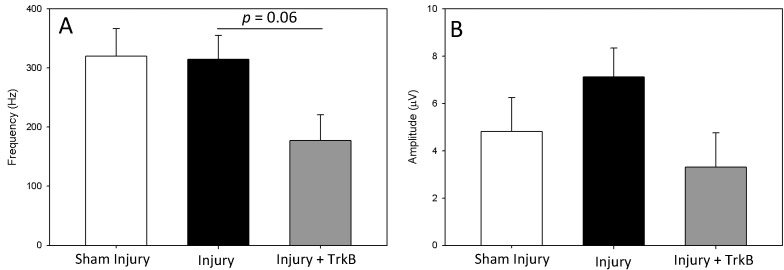
External urethral sphincter electromyography results. External urethral sphincter (EUS) electromyography (EMG) firing rate (**A**) and amplitude (**B**) 6 weeks after sham injury treated with saline (Sham Injury), pudendal nerve crush (PNC) and vaginal distension (VD) injury treated with saline (Injury), or PNC + VD injury treated with tyrosine kinase B (TrkB) fusion-chimera (Injury + TrkB). Each bar represents mean ± standard error of the data from 11–14 animals. There were 4–6 EUS EMG tests per animal. A Welsh one-way analysis of variance (ANOVA) with a Dunnett’s T3 post hoc test was used to determine statistically significant differences (*p* < 0.05) compared to the Injury group.

**Figure 4 ijms-24-04998-f004:**
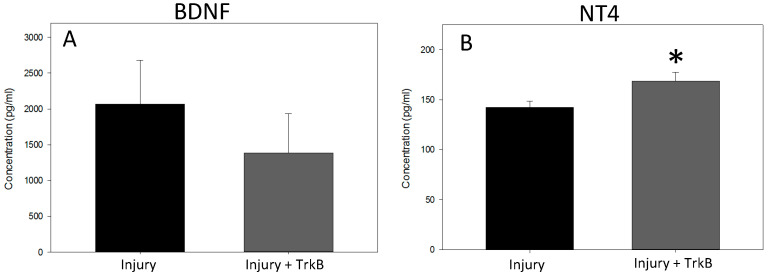
Effects of Tyrosine Kinase B functional chimera binding. Plasma concentration of unbound (**A**) brain-derived neurotrophic factor (BDNF) and (**B**) neurotrophin 4 (NT4) six weeks after pudendal nerve crush (PNC) and vaginal distension (VD) injury treated with saline (Injury), or PNC + VD injury treated with tyrosine kinase B (TrkB) fusion chimera (Injury + TrkB). Each bar represents the mean ± standard error of the mean of data from 8–12 animals in each group. A Mann–Whitney test was used to determine statistically significant differences. * indicates a statistically significant difference (*p* < 0.05) between the groups.

**Figure 5 ijms-24-04998-f005:**
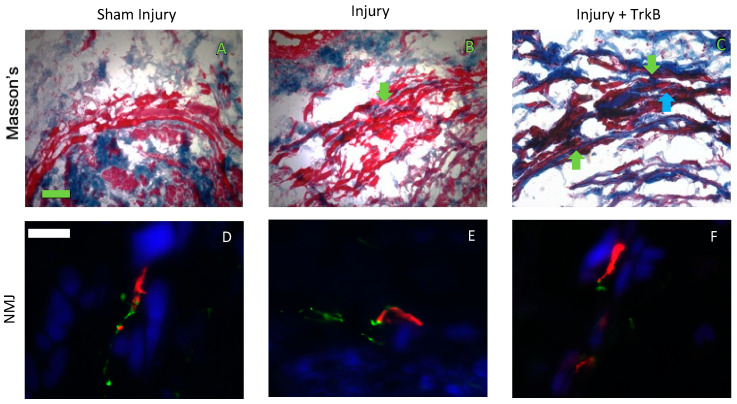
External urethral sphincter anatomy. Representative images of Masson’s stained urethral cross-sections 6 weeks after (**A**) sham dual injury treated with saline (Sham Injury); (**B**) pudendal nerve crush (PNC) and vaginal distension (VD) injury treated with saline (Injury); and (**C**) PNC + VD injury treated with tyrosine kinase B (TrkB) fusion-chimera (Injury + TrkB). Representative images of neuromuscular junction (NMJ) immunofluorescence 6 weeks after (**D**) sham injury, (**E**) injury, and (**F**) injury + TrkB. In Masson’s stained specimens (**A**–**C**), collagen was stained blue, and muscle cells were stained red. In immunofluorescence images (**D**–**F**), nerve axons are stained green, NMJs are stained red, and the muscle is stained blue. The green scale bar in panel (**A**) for Masson’s trichrome data is 100 µm. The white scale bar in panel (**D**) for immunofluorescence data is 20 µm. Green arrows indicate collagen, and the blue arrow indicates disrupted EUS.

**Figure 6 ijms-24-04998-f006:**
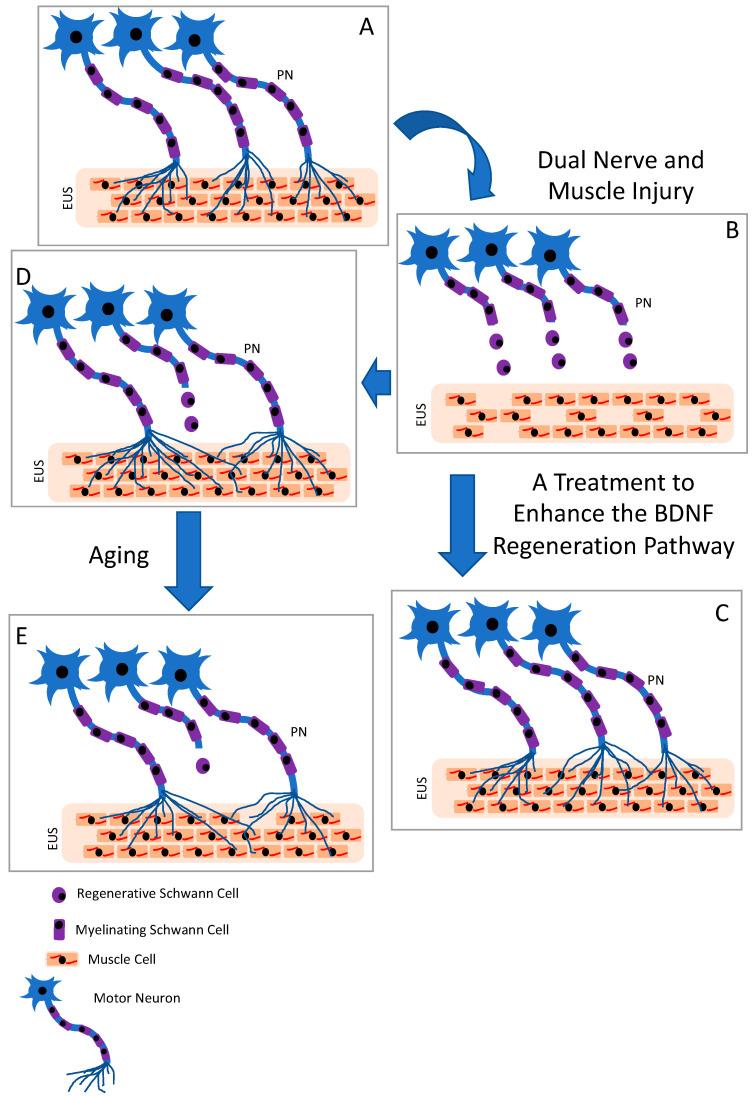
Summary diagram. A normal external urethral sphincter (EUS) is innervated by the motor neurons of the pudendal nerve (PN), as shown in (**A**). An injured EUS and PN has decreased innervation, as shown in (**B**). Treatments that enhance the BDNF regenerative pathway could enhance neuroregeneration of the PN and re-innervation of the EUS, as shown in (**C**). Unassisted regeneration of the PN results in increase in muscle fibers innervated by a single motor neuron, as shown in (**D**). With time after unassisted partial regeneration, axons atrophy with aging, as shown in (**E**).

## Data Availability

The work was funded by the Department of Veteran’s Affairs and the research was conducted at the Louis Stokes Veteran’s Affairs Medical Center and is covered under the Freedom of Information Act. Any request for information must be sent to the Department of Veteran’s Affairs and all requests will be honored. Requests to access the datasets should be directed to website: https://www.va.gov/foia/, email: vacofoiase@va.gov.

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
