# Peer review of "Brain-Derived Neurotrophic Factor Is Indispensable to Continence Recovery after a Dual Nerve and Muscle Childbirth Injury Model"

_ijms, 2023, doi:10.3390/ijms24054998_

Round 1

Reviewer 1 Report

The present study, "Brain-Derived Neurotrophic Factor is Indispensable to Continence Recovery after a Dual Nerve and Muscle Childbirth Injury Model", aimed to use tyrosine kinase B (TrkB), the receptor of BDNF, to bind free BDNF and inhibit spontaneous regeneration in a rat model of SUI.

The study not only met its goal, demonstrating that BDNF could be used for neuromuscular injuries caused by childbirth, but could also be used as a therapeutic agent for the recovery of other traumatic and non-traumatic nerve conditions involving neuromotor disturbances. Furthermore, this information could be taken into account for studies involving neurodegeneration from various causes. I think the study should be accepted in its current form.

Author Response

We thank the reviewers for the constructive comments. We have edited the manuscript in accordance with the comments, improving it as a result. Our point by point response to the critiques are in red font below.

Reviewer 1. The present study, "Brain-Derived Neurotrophic Factor is Indispensable to Continence Recovery after a Dual Nerve and Muscle Childbirth Injury Model", aimed to use tyrosine kinase B (TrkB), the receptor of BDNF, to bind free BDNF and inhibit spontaneous regeneration in a rat model of SUI.

The study not only met its goal, demonstrating that BDNF could be used for neuromuscular injuries caused by childbirth, but could also be used as a therapeutic agent for the recovery of other traumatic and non-traumatic nerve conditions involving neuromotor disturbances. Furthermore, this information could be taken into account for studies involving neurodegeneration from various causes. I think the study should be accepted in its current form.

Thank you for your positive review.

Reviewer 2 Report

Summary

            This study evaluates the effect of decreased brain-derived neurotropic factor (BDNF) in dual nerve and muscle injuries that correspond with stress urinary incontinence. The authors used a dual-nerve injury mouse model where the mice were anesthetized and the pudental nerve was surgically visualized and crushed twice with forceps. The sham group had the pudental nerve exposed, but not crushed. An osmotic pump was then placed that delivered either saline or TrkB-Fc in saline to bind and inhibit endogenous brain-derived neurotropic factor (BDNF), and the incision closed. The second injury utilized a foley catheter to inflate the vagina with 3 mL of water for four hours (vaginal distention, VD) and then deflated and removed it. Animals with the sham VD had the catheter inserted, but no inflation of the vagina. Six weeks later, the authors evaluated for pudental nerve motor branch function, and then simultaneously recorded external urethral sphincter (EUS) electromyography (EMG) recording and leak point pressure (LPP) testing in three groups 1. Sham PNC + VD + Saline, 2. PNC + VD + Saline, and 3. PNF + VD + TrkB-Fc/saline. The authors hypothesize that BDNF is required for functional recovery, testing that through utilizing the TrkB-Fc receptor to bind endogenous BDNF. A previous study has added BDNF in and shown that improved healing time, and this study attempts to decrease endogenous BDNF and shows slowed healing time and suggests that decreased BDNF would result in an increased number of muscle fibers, which may maie it more susceptible to damage and recurrent stress urinary incontinence. Overall, the paper is well written, but significant revisions and potentially additional experiments are necessary for the manuscript to be of publication quality.

Major Points

1.     Why was the 12 microgram/day of the TrkB-Fc dose selected? Did the authors perform a dose-response curve to select the appropriate dose? If no, why not?

2.     Figure 1. How many animals were used per group? How many times was the experiment replicated?

3.     Is Figure 2 all the data combined for the bottom three panels on Figure 1? Or was it measured separately?

4.     Is Figure 3 all the data combined for the top three panels on Figure 1? Or was it measured separately.

5.     Figure 4A. There is no statistically significant difference between BDNF amounts in the Injury vs. the Injury + TrkB model at the time point shown.

a.     What amount of downregulation is necessary for biological significance? Is that the same as statistical significance in this model?

b.     Did the authors perform the analysis to show at any time-point that the BDNF amounts were statistically significantly different between the groups?

c.     The authors postulate that BDNF and NT4 may have been upregulated at 6 weeks, but do not show data to support it. As currently written, without experimental data showing the BDNF was downregulated at some point, I’m not convinced the authors data supports the current title of the manuscript. What if there were issues with the quality of the reagent used?

6.     Figure 5 how many times was this assay performed? Sections from how many animals were used for each group? Was the innervation patterns quantified? If not, then the data doesn’t appear to support Figure 6 Diagrams D & E

Minor Points

1.     Figure 4. Please note on the panels that A is BNDF and B is NT4. I realize it is stated in the figure legend, but will make the figure more clear.

Author Response

We thank the reviewers for the constructive comments. We have edited the manuscript in accordance with the comments, improving it as a result. Our point by point response to the critiques are in red font below.

Thank you for your review and comments. The revisions we have made in response to the comments have improved the manuscript.

Major Points

  1. Why was the 12 microgram/day of the TrkB-Fc dose selected? Did the authors perform a dose-response curve to select the appropriate dose? If no, why not?

Thank you for your questions. The dose of 12 microgram/day was chosen because previous studies have used it without any unintended side effects. We did not perform a dose-response curve, because of the previously publication. We have added these details to the Introduction and Methods sections. Additionally, we have added a statement to the limitations paragraph in the Discussion section.

  1. Figure 1. How many animals were used per group? How many times was the experiment replicated?

Thank you for the comment. We used 11-14 animals per group, and functional testing was repeated in each animal 4-6 times. In the final result, there were 12 animals in the sham injury group, 14 in the injury with saline treatment, and 11 in the injury with TrkB treated group. We have added this information to the Methods section and relevant Figure Legends.

  1. Is Figure 2 all the data combined for the bottom three panels on Figure 1? Or was it measured separately?

Thank you for your question. The bottom panel of Figure 1 is a representative pressure traces during LPP testing, the mean and standard error of the mean are graphed in Figure 2. Each animal was tested individually, and LPP and EUS EMG were recorded simultaneously. Representative traces consist of LPP and the corresponding EUS EMG trace (Figure 1), which were chosen because they were the closest to the mean of both outcomes per group.  We have added this information to the Figure Legends of Figures 1 and 2 and to the Methods section.

  1. Is Figure 3 all the data combined for the top three panels on Figure 1? Or was it measured separately.

Thank you for your comment. We have revised the figure to remove the middle set of traces, which was included in error. The top traces are example EUS EMG traces. The mean and standard error of the mean are graphed in Figure 3. Each animal was tested individually, and LPP and EUS EMG were recorded simultaneously.  Representative traces are the LPP pressure and corresponding EUS EMG trace (Figure 1), which were chosen because they were the closest to the mean of both outcomes per group. We have added this information to the Figure Legend of Figure 1 and to the Methods section. We have revised Figure 1 and added it to the file.

  1. Figure 4A. There is no statistically significant difference between BDNF amounts in the Injury vs. the Injury + TrkB model at the time point shown.     A.What amount of downregulation is necessary for biological significance? Is that the same as statistical significance in this model?

Thank you for your comment. While we don’t know the smallest reduction needed to get a biological effect, we previously showed that after 1 week of treatment at this dose, TrkB-Fc significantly inhibited the BDNF regeneration pathway, indicated by a decrease in bII tubulin expression. This information has been added to the Discussion section. Others have seen a biological effect (in culture) with as little as 25% reduction in BDNF, suggesting that a 45% reduction for at least the first three weeks would reduce regeneration in this system. Both the 1 week and 3 weeks studies were published in Balog et al. 2020. This manuscript looked at the effects of TrkB-Fc treatment on regeneration and functional recovery after pudendal nerve crush. This information has been clarified in the Discussion section.

B. Did the authors perform the analysis to show at any time-point that the BDNF amounts were statistically significantly different between the groups?

Thank you for your comment. We have previously shown that with 3 weeks of TrkB-Fc treatment free BDNF was reduced by ~45%, a statistically significant reduction which showed a reduction in functional recovery. This information has been clarified in the Discussion section.

C. The authors postulate that BDNF and NT4 may have been upregulated at 6 weeks, but do not show data to support it. As currently written, without experimental data showing the BDNF was downregulated at some point, I’m not convinced the authors data supports the current title of the manuscript. What if there were issues with the quality of the reagent used?

Thank you for your comment. We used the same brand of osmotic pumps and the same company and batch for the TrkB-Fc between this study and the experiments described in Balog et al. 2020 and 2021. It is a limitation that we did not test the function of the TrkB-Fc at the end of the six weeks, which means the TrkB-Fc may have not been active for the last three weeks of the study. We have added this to the Discussion section.

  1. Figure 5 how many times was this assay performed? Sections from how many animals were used for each group? Was the innervation patterns quantified? If not, then the data doesn’t appear to support Figure 6 Diagrams D & E

Thank you for your question. Four to five animals were analyzed per group, using one slide per animal. This information was added to the Methods section. The innervation patterns were qualitatively assessed based on whether the NMJs were innervated or not and if it was innervated by a single axon or multiple, as we have done before. We have added references to the Methods section.

Figure 6D were based on a previous publication.  Song et al. 2015, showed that while LPP had recovered 9 weeks after a PNC + VD, some of the NMJs were still impaired, with diffuse acetylcholine receptors. We have added this to the Discussion section and the references to the reference list. Panel E is a hypothesized mechanism for development of late in life SUI that was not tested in this study.  

Minor Points

  1. Figure 4. Please note on the panels that A is BNDF and B is NT4. I realize it is stated in the figure legend, but will make the figure more clear.

Thank you for your comment. We have revised the figure and added it to manuscript.

Round 2

Reviewer 2 Report

The authors have sufficiently addressed my concerns and revised the manuscript accordingly to now warrant publication in IJMS. Thank you.